# Lie Point Symmetry and Physics Informed Networks

**Tara Akhound-Sadegh** *
School of Computer Science, McGill University,
Mila - Quebec Artificial Intelligence Institute,
Montreal, Quebec, Canada

**Laurence Perreault-Levasseur**
Université de Montréal, Montreal, Quebec, Canada
Ciela Institute, Montreal, Quebec, Canada
Mila - Quebec Artificial Intelligence Institute, Montreal, Quebec, Canada
Trottier Space Institute, Montreal, Quebec, Canada
CCA, Flatiron Institute, New York, USA
Perimeter Institute, Waterloo, Ontario, Canada

**Johannes Brandstetter**
Microsoft Research AI4Science,
Amsterdam, Netherlands

**Max Welling**
University of Amsterdam, †
Amsterdam, Netherlands

**Siamak Ravanbakhsh**
School of Computer Science, McGill University,
Mila - Quebec Artificial Intelligence Institute,
Montreal, Quebec, Canada

## Abstract

Symmetries have been leveraged to improve the generalization of neural networks
through different mechanisms from data augmentation to equivariant architectures.
However, despite their potential, their integration into neural solvers for partial
differential equations (PDEs) remains largely unexplored. We explore the inte-
gration of PDE symmetries, known as Lie point symmetries, in a major family of
neural solvers known as physics-informed neural networks (PINNs). We propose
a loss function that informs the network about Lie point symmetries in the same
way that PINN models try to enforce the underlying PDE through a loss function.
Intuitively, our symmetry loss ensures that the infinitesimal generators of the Lie
group conserve the PDE solutions. Effectively, this means that once the network
learns a solution, it also learns the neighbouring solutions generated by Lie point
symmetries. Empirical evaluations indicate that the inductive bias introduced by the
Lie point symmetries of the PDEs greatly boosts the sample efficiency of PINNs.

## 1   Introduction

In recent years, deep learning has accelerated data-driven approaches to science and
engineering.    A prominent example is the role of deep learning in solving partial
differential  equations  (PDEs),  which  are  ubiquitous  in  many  scientific  disciplines.

*Corresponding author: `tara.akhound-sadegh@mila.quebec`
†Contribution was done when at Microsoft Research AI4Science, Amsterdam.

37th Conference on Neural Information Processing Systems (NeurIPS 2023).

This is mainly driven by the fact that traditional handcrafted numerical solvers can be prohibitively expensive, and thus, learning to solve PDEs has the potential to significantly impact various areas of science, ranging from quantum chemistry to biology to climate science to cosmology, [Wang et al., 2020a, Kochkov et al., 2021, Kashinath et al., 2021, Gupta and Brandstetter, 2022, Nguyen et al., 2023, Bi et al., 2022]. Like any other machine learning problem, learning to solve PDEs can benefit from inductive biases to boost sample efficiency and generalization capabilities. As such, the PDE itself and its respective Lie point symmetries, i.e. joint symmetries of coordinate and field transformations, are two natural inductive biases for constructing efficient neural PDE solvers.

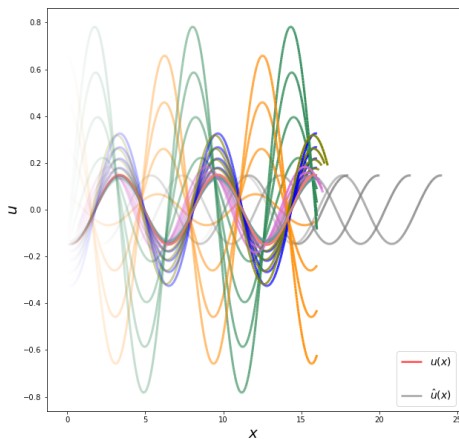

Figure 1: All the solutions of the differential equation of a harmonic oscillator can be reached via symmetry transformations of a single solution. Its Lie-point symmetry group is the special linear group, $SL(3)$, corresponding to eight one-parameter subgroups transforming a given solution, identified using a different colour in this figure. While, in general, Lie-point symmetries form more than one orbit – that is, we cannot reach all solutions using such transformations – they can significantly narrow down the solution space.

For parametric PDEs, Physics-Informed Neural Networks (PINNs) have emerged as an efficient new learning paradigm. PINNs constrain the solution network to satisfy the underlying PDE via the PINN loss terms [Raissi et al., 2019], which comprise a residual loss, as well as loss terms for initial and boundary points. As such, PINNs can be seen as a data-free learning approach, utilizing available physics information to guide neural network training. Lie point symmetries are uniquely special in that it is possible to characterize the full set of permissible transformations for each PDE, and thus naturally offer additional physics information which piggy-backs the data-free learning objective. Lie point symmetries have been introduced to deep learning by Brandstetter et al. [2022a] for Lie point symmetry data augmentation which places a firm mathematical footing on data augmentation pipelines for neural PDE solvers. Further, Lie point symmetries showed promising results when used in extending the framework to self-supervised learning for neural PDE solvers [Mialon et al., 2023]. In this work, for the first time we integrate symmetries into PINNs and demonstrate the effectiveness of this symmetry regularization on generalization capabilities.

Lie point symmetries of a PDE, by definition, map a solution to a solution, preserving the PDE. Lie point symmetry transformations act on both independent (*e.g.*, time and space), and dependent (PDE solution) fields and, by extension, on all partial derivatives. To enforce a Lie point symmetry, we first need to *prolong* a symmetry transformation to find how it transforms the partial derivatives of the respective PDE. We show how to do this using automatic differentiation, which enables us to transform the entire PDE for each infinitesimal generator of each Lie point symmetry. "Imposing a symmetry" means that by transforming the PDE via a symmetry transformation, the PDE is required to be preserved. In practical terms, Lie point symmetries should be orthogonal to the PDE gradient. Each Lie Point symmetry results in one such orthogonality constraint, which we enforce using a penalty term within the PINN framework. We can enforce these on the same points where the PINN loss is imposed. Importantly, our Lie Point symmetry regularization is complementary to the PINN loss: while the PDE loss regularizes the neural network to satisfy the PDE at select points, the Lie point symmetry loss regularizes the neural network such that infinitesimal changes in different directions continue to satisfy the underlying PDE.

## 2 Background

### 2.1 Partial Differential Equations

Partial differential equations (PDEs) are used to mathematically describe the dynamics of various physical systems. In a PDE, the evolution of a function that involves several variables is described in terms of local updates expressed by partial derivatives. Since temporal PDEs are especially prevalent

in physical sciences, we consider PDEs of the following general form:

$$\Delta = \mathbf{u}_t + \mathcal{D}_\mathbf{x}[\mathbf{u}] = 0, \qquad\qquad t \in [0, T], \mathbf{x} \in \Omega \qquad (1)$$
$$\mathbf{u}(0, \mathbf{x}) = f(\mathbf{x}), \qquad\qquad \mathbf{x} \in \Omega$$
$$\mathbf{u}(t, \mathbf{x}) = g(t, \mathbf{x}), \qquad\qquad \mathbf{x} \in \partial\Omega, t \in [0, T]$$

where $\mathbf{u}(t, \mathbf{x}) \in \mathbb{R}^{D_u}$ is the solution to the PDE, $t$ denotes the time, and $\mathbf{x}$ is a vector of possibly multiple spatial coordinates. $\Omega \subset \mathbb{R}^D$ is the domain and $\mathcal{D}_\mathbf{x}[.]$ is a non-linear *differential operator*. $f(\mathbf{x})$ is known as the initial condition function and $g(t, \mathbf{x})$ describes the boundary conditions. In the following, we sometimes explicitly separate the time from other independent variables and sometimes for convenience group them together.

**Example 1** (Heat Equation). *The one-dimensional heat equation describes the heat conduction in a one-dimensional rod, with viscosity coefficient $\nu$. Its differential operator is given by $\mathcal{D}_x[u] = -\nu u_{xx}$.*

**Using Neural Networks for Solving PDEs.** For many systems, obtaining an analytical solution to the PDE is impossible; hence, numerical methods are traditionally used to obtain approximate solutions. Numerical solvers, such as finite element methods (FEM) or finite difference methods (FDM) rely on discretizing the space $T \times \Omega$ [Quarteroni, 2009]. The topology of the space has to be taken into account when constructing the mesh, and the resolution of the discretization will affect the accuracy of the predicted solutions. Additionally, these solvers are often computationally expensive, especially for complex dynamics, and each time the initial or boundary conditions of the PDE change the solver must be rerun. These considerations and constraints make designing numerical solvers difficult, and scientists often need to handcraft a specific solver for an application [Quarteroni, 2009].

Given the recent successes of neural networks, especially in dealing with large datasets, using deep learning to solve PDEs has become a promising direction. The idea is to learn a function or, more generally, a functional that bypasses the numerical computation and produces the solution to the PDE in a single shot or by progression through time. Broadly, there are two approaches to solving PDEs with neural networks: neural operator methods which approximate the solution operator that maps between solutions of the underlying PDEs, and direct methods which learn the underlying solution function. The most prominent representative of the latter are PINNs.

**PINNs.** In contrast to neural operator methods [Lu et al., 2021, Li et al., 2021a] where the main idea is to generalize neural networks to obtain mappings between function space, PINNs directly learn the solution function of the underlying PDE. Consequently instead of relying on large training sets,

PINNs operate as a surrogate model for the PDE solution, trained directly with the PDE equation itself. The simplicity of the PINN idea has made it the subject of many follow-up improvements; see Krishnapriyan et al. [2021], Wang et al. [2020b, 2022, 2023]

In PINNs, the PDE solution $u(t, \mathbf{x})$ of Eq. (1) is a neural network $u_\theta(t, \mathbf{x})$ with parameters $\theta$. The loss function is then compromised of two parts:

$$\mathcal{L}(\theta) = \mathcal{L}_{\text{PDE}} + \mathcal{L}_{\text{data-fit}} \qquad (2)$$

The first term is the *physics-informed* objective, ensuring that the function learned by the neural network satisfies the PDE Eq. (1). Let $r(t, \mathbf{x})$ denote the residual error in agreement with the PDE equation:

$$r_\theta(t, \mathbf{x}) = \frac{\partial}{\partial t} u_\theta(t, \mathbf{x}) + \mathcal{D}_\mathbf{x}[u_\theta(t, \mathbf{x})]$$

where the derivatives of the solution network $u_\theta$ are calculated using automatic differentiation [Raissi et al., 2019]. This penalty is then imposed on a finite set of points $(t, \mathbf{x})_{1:N_r}$ which are sampled from inside the domain $[0, T] \times \Omega$ to obtain the *PDE loss*:

$$\mathcal{L}_{\text{PDE}} = \frac{1}{N_r} \sum_{i=1}^{N_r} \|r_\theta(t_i, \mathbf{x}_i)\|_2^2 \qquad (3)$$

The second term in Eq. (2), is a supervised loss which ensures that the function learned by the neural network satisfies the initial and boundary conditions of the problem – that is

$$\mathcal{L}_{\text{data-fit}} = \frac{1}{N_s} \sum_{i=1}^{N_s} \|u_\theta(0, \mathbf{x}_i^0) - f(x_i^0)\|_2^2 + \frac{1}{N_b} \sum_{i=1}^{N_b} \|u_\theta(t_i^b, \mathbf{x}_i^b) - g(t_i^b, x_i^b)\|_2^2 \qquad (4)$$

where $(\mathbf{x}^0)_{1:N_s} \in \Omega$ are $N_s$ samples at which the initial condition function $f$ is sampled. $(t^b, \mathbf{x}^b)_{1:N_b}$ are $N_b$ points sampled on the boundary (from $[0, T] \times \partial\Omega$).

There have also been multiple models that combine the operator learning approach with the physics-informed loss. For example, in Li et al. [2021b], the NO approach is combined with the PINN loss. Another example is Wang et al. [2021], which we describe further in Section 3.1.

## 2.2 Symmetries

**Groups.** Symmetries are essentially transformations of the object that leave an aspect of it invariant: for example, the nature of an object does not change if we translate or rotate it. In mathematics, symmetries of an object are described by abstract objects known as *groups*. More concretely, a group is a set $G$ and a group operation $\cdot : G \times G \to G$ satisfying: associativity, the existence of an identity element, and the presence of an inverse element for every element in the set.

**Lie Groups.** Groups that are also differentiable manifolds are known as *Lie groups*, and are important in the study of continuous symmetries. In Lie groups, in addition to satisfying the three properties, the group operation $\cdot$ and its inverse are smooth maps. Each Lie group has an associated *Lie algebra* which is its tangent space, as a vector space, at identity. [3] Intuitively, Lie algebras describe the smooth transformations of the Lie groups in the limit of infinitesimal transformations. The elements of the Lie algebra are vectors describing the direction of the infinitesimal symmetry transformation.

**One Parameter Subgroups.** The Lie group, $\mathcal{G}$, can be multi-dimensional and complex. When the $n$-dimensional group $\mathcal{G}$ is *simply connected*, it is often represented in terms of a series of $n$ *one-parameter transformations*, $g = g_1(\epsilon_1)g_2(\epsilon_2)\ldots g_n(\epsilon_n)$, where $g_i : \mathbb{R} \to \mathcal{G}$, $i = 1, \ldots, n$ and such that $g_i(\epsilon)g_i(\delta) = g_i(\epsilon + \delta)$. The $\epsilon$'s are the real parameters of the transformation, and each $g_i$ is a continuous group homomorphism (*i.e.*, a smooth, group-structured map) from this parameter to the symmetry group.

## 2.3 Symmetries of Partial Differential Equations

In this section, we will provide the mathematical background of symmetries of differential equations. We will mostly follow the exposition presented in Olver [1986] and refer the reader to this original text for a more in-depth treatment of this topic.

In the context of differential equations, symmetries are transformations that map a solution of the PDE to another solution. For example, in Fig. 1, we can see how the solutions of a simple harmonic oscillator can be obtained via symmetry transformations of a given solution, where the symmetry group is $SL(3)$. Consider the PDE $\Delta$, involving $p$ independent variables $\mathbf{x} = (x_1, \ldots, x_p) \in X$ and $q$ dependent variables $\mathbf{u} = (u_1, \ldots, u_q) \in U$. The solutions to the PDE will be of the form $u_i = f_i(x_1, \ldots, x_p)$ for $i = 1, \ldots, q$, for $\mathbf{x} \in \Omega$, where $\Omega \subset X$ is the domain of the function f. The symmetry group $\mathcal{G}$, of $\Delta$, is the local group of transformations on an open subset of the space of dependent and independent variables, $M \subset X \times U$, transforming solutions of $\Delta$ to other solutions.

**Prolongations.** To formalize this abstract definition of PDE symmetries, Lie proposed viewing $\Delta$ as a concrete geometric object and introduced the concept of *prolongation* [Olver, 1986]. The idea is to *prolong* the space of independent and dependent variables, $X \times U$, to a space that also represents the partial derivatives involved in the PDE. More concretely, we have the following definition:

**Definition 1.** *The **n-th order prolongation** (or n-th order* jet space*) of $X \times U$ is denoted as $X \times U^{(n)} = X \times U_1 \times \cdots \times U_n$, whose coordinates represent the independent and dependent variables as well as all the partial derivatives of the dependent variables up to order n.*

Equivalently we have the notion of *prolongation* of $\mathbf{u}$ as $\mathbf{u}^{(n)} = (\mathbf{u_x}, \mathbf{u_{xx}}, \ldots, \mathbf{u_{nx}})$ where $\mathbf{u}_{i\mathbf{x}}$ is all the unique $i^{\text{th}}$ derivatives of $u$, for $i = 1, \ldots, n$. For example, if $\mathbf{x} = (x, y)$, then $\mathbf{u_{xx}} = (\partial_{xx}\mathbf{u}, \partial_{xy}\mathbf{u}, \partial_{yy}\mathbf{u})$.

---

[3]Formally, the Lie algebra is a vector space equipped with a binary operation known as the Lie bracket.

Using this notion of prolongation, we can represent a PDE as an algebraic equation, $\Delta(\mathbf{x}, \mathbf{u}^{(n)}) = 0$, where $\Delta$ is the map that determines the PDE, *i.e.*, $\Delta : X \times U^{(n)} \to \mathbb{R}$. In other words, the PDE tells us where the map $\Delta$ vanishes on $X \times U^{(n)}$.

For example, for the heat equation described in 1, $\Delta$ is given by:

$$\Delta((x, t), \mathbf{u}^{(2)}) = u_t - \nu u_{xx} , \tag{5}$$

The graph of all prolonged solutions is the set $\mathcal{S}_\Delta \subset X \times U^{(n)}$, and is defined as $\mathcal{S}_\Delta = \{(\mathbf{x}, \mathbf{u}^{(n)}) : \Delta(\mathbf{x}, \mathbf{u}^{(n)}) = 0\}$. In this new notation, we can say that $\mathbf{u}(\mathbf{x})$ is a solution of the PDE if $\Gamma_u^{(n)} = \{(\mathbf{x}, \mathrm{pr}^{(n)}\mathbf{u}(\mathbf{x}))\} \subset \mathcal{S}_\Delta$. where $\mathrm{pr}^{(n)}\mathbf{u}(\mathbf{x}) : X \to U^n$, is a vector-valued function whose entries represent all derivatives of $\mathbf{u}$ wrt $\mathbf{x}$ up to order $n$.

**Prolongations of the Infinitesimal Generators.** Let $\mathbf{v}$ be the vector field on the subspace $M \subset X \times U$ with corresponding one-parameter subgroup $\exp(\epsilon\mathbf{v})$. In other words, the vector field $\mathbf{v}$ is the *infinitesimal generator* of the one-parameter subgroup. Intuitively, this vector field describes the infinitesimal transformations of the group to the independent and dependent variables, and we can write it as:

$$\mathbf{v} = \sum_{i=1}^{p} \xi_i(\mathbf{x}, \mathbf{u})\frac{\partial}{\partial x^i} + \sum_{\alpha=1}^{q} \phi_\alpha(\mathbf{x}, \mathbf{u})\frac{\partial}{\partial u^\alpha} , \tag{6}$$

where $\xi^i(\mathbf{x}, \mathbf{u})$ and $\phi_\alpha(\mathbf{x}, \mathbf{u})$ are coordinate-dependent coefficients. To study how symmetries transform a solution to another solution, we need to know how they transform the partial derivatives and, therefore, the *jet space* $X \times U^{(n)}$.

A symmetry transformation of the independent ($\mathbf{x}$) and dependent ($\mathbf{u}$) variables will also induce transformations in the partial derivatives $\mathbf{u_x}, \mathbf{u_{xx}}, \ldots$. The *prolongation of the infinitesimal generator*, $\mathrm{pr}^{(n)}\mathbf{v}$ is a generalization of the generator $\mathbf{v}$ which describes these induced transformations in these partial derivatives.

This prolongation will be defined on the jet-space $X \times U^{(n)}$ and it is given by:

$$\mathrm{pr}^{(n)}\mathbf{v} = \sum_{i=1}^{p} \xi_i(\mathbf{x}, \mathbf{u})\frac{\partial}{\partial x^i} + \sum_{\alpha=1}^{q} \sum_{J} \phi_\alpha^{(J)}(\mathbf{x}, \mathbf{u})\frac{\partial}{\partial u_J^\alpha} , \tag{7}$$

where we have used the notation $J = (i_1, \ldots, i_k)$ for the multi-indices, with $0 \leq i_k \leq p$ and $0 \leq k \leq n$ and $\mathbf{u}_J^\alpha = \frac{\partial^k \mathbf{u}^\alpha}{\partial x^{i_1}..\partial x^{i_k}}$. Calculating $\phi_\alpha^{(J)}$, the coefficients of $\partial_{\mathbf{u}_J^\alpha}$, can be done using the prolongation formula, which involves the total derivative operator $D$ (see Olver [1986] for a derivation):

$$\phi_\alpha^{(J)} = D_J Q_\alpha + \sum_{i=1}^{p} \xi_i \frac{\partial \mathbf{u}_J^\alpha}{\partial x^i} \qquad \text{where} \qquad Q_\alpha = \phi_\alpha - \sum_{i=1}^{p} \xi_i \frac{\partial \mathbf{u}^\alpha}{\partial x^i} . \tag{8}$$

The upshot is that we can mechanically calculate the prolonged vector field using partial derivatives of $\mathbf{u}$, which are, in turn, produced by automatic differentiation. In practice, prolonged vector fields are implemented as vector-valued functions (or functionals) of $\mathbf{x}$ and $\mathbf{u}$. Therefore, the implementation of this mechanical process is generic and can be applied to any PDE. See the Appendix A for examples.

**Lie Point Symmetries of PDEs.** We can now define the *prolongation of the action of* $\mathcal{G}$:

**Definition 2.** *For symmetry group $\mathcal{G}$ acting on $M$, the **prolongation of action of** $\mathcal{G}$ on the open subset $M \subset X \times U$ is the induced action on $M^{(n)} = M \times U^{(n)}$ which transforms derivatives of a solution, $\mathbf{u} = f(\mathbf{x})$, into corresponding derivatives of another solution, $\mathbf{u}' = f'(\mathbf{x}')$. We can write this as:*

$$\mathrm{pr}^{(n)}g \cdot (\mathbf{x}, \mathbf{u}^{(n)}) = (g \cdot \mathbf{x}, \mathrm{pr}^{(n)}g \cdot \mathbf{u}^{(n)}) .$$

Using the definition above, we can provide a new criterion for $\mathcal{G}$ being a symmetry group of $\Delta$, under a mild assumption on the PDE equation.[4]

---

[4]This assumption is that $\Delta$ is of maximal rank. We refer the readers to Olver [1986] for the definition of this condition. However, we note that this assumption does not pose a restriction since for any PDE not satisfying this condition, it is possible to find an equivalent PDE which does.

**Theorem 2.1.** $\mathcal{G}$ is the symmetry group of the $n$-th order PDE $\Delta(\mathbf{x}, \mathbf{u}^n)$, if $\mathcal{G}$ acts on $M$, and its prolongation leaves the solution set $\mathcal{S}_\Delta$ invariant: $\mathrm{pr}^{(n)} g \cdot (\mathbf{x}, \mathbf{u}^{(n)}) \in \mathcal{S}_\Delta, \quad \forall g \in \mathcal{G}$.

Finally, we can express the symmetry condition in terms of the infinitesimal generators $\mathbf{v}$ of $\mathcal{G}$:

**Theorem 2.2** (Infinitesimal Criterion). $\mathcal{G}$ is a symmetry group of the PDE $\Delta(\mathbf{x}, \mathbf{u}^n)$ if for every infinitesimal generator $\mathbf{v}$ of $\mathcal{G}$, we have that

$$\mathrm{pr}^{(n)} \mathbf{v}[\Delta] = 0 \quad \text{when} \quad \Delta = 0 .$$

**Example 2** (A Symmetry of the Heat Equation). *As an illustrative example, we can consider the heat Eq. (5) and will show that the following vector field generates a symmetry group for this PDE:* $\mathbf{v} = 2\nu t \partial_x - xu \partial_u$. *We need to find the first prolongation $\phi^{(t)}$ and the second prolongation $\phi^{(xx)}$, where $\phi = -xu$. Using the prolongation formula given in Eq. (7), we get:*

$$\phi^{(t)} = -xu_t - 2\nu u_x \qquad and \qquad \phi^{(xx)} = -2u_x - xu_{xx}$$

*Now:* $\mathrm{pr}^{(2)} \mathbf{v}[\Delta] = \phi^{(t)} - \nu \phi^{(xx)} = xu_t + 2\nu u_x - \nu\big(2\nu u_x - x\nu u_{xx}\big) = x(u_t - \nu u_{xx})$ .

*Clearly $\mathrm{pr}^{(2)} \mathbf{v}[\Delta] = 0$ when $\Delta = 0$, hence $\mathbf{v}$ is a symmetry of the heat equation.*

## 3 Methods

### 3.1 Solving PDEs with Different Initial/Boundary Conditions with PINNs

Wang et al. [2021] combines the approach introduced in Lu et al. [2021] with the PINN loss to solve PDEs with different initial or boundary conditions without requiring retraining of the model as the original PINN model does. We will also use this proposed framework to examine the effect of enforcing the symmetry condition of the PDE on the model.

Recall that we want to model the operator $\mathcal{O} : \mathcal{A} \rightarrow \mathcal{U}$, where $\mathcal{A}$ is the space of initial condition functions and $\mathcal{U}$ is the space of PDE solutions. Our model consists of two neural networks: $e_{\theta_1}$ embeds the initial condition function, and $g_{\theta_2}$ embeds the independent variables, $[t, \mathbf{x}] \in \mathbb{R}^p$.

In particular, to embed the initial condition function $f(\mathbf{x}) = \mathbf{u}(0, \mathbf{x}) \in \mathbb{R}^p$, it is sampled at fixed points $\{\mathbf{x}_1, \ldots, \mathbf{x}_n\}$, and the concatenated values are fed to $f_{\theta_1}$

The final prediction is the inner product of these embedding vectors:

$$\mathcal{O}_\theta(f)(\mathbf{x}, t) = e_{\theta_1}^\top \big(\mathbf{u}(0, \mathbf{x}_1), \ldots, \mathbf{u}(0, \mathbf{x}_n)\big) \, g_{\theta_2}(\mathbf{x}, t) , \tag{9}$$

where $\theta = (\theta_1, \theta_2)$. In Algorithm 1, we use the notation $u_\theta$ to denote the operator $\mathcal{O}_\theta$ and use $(\mathbf{x}l, \mathbf{u}^l)_{1:N_l}$ to include both boundary and initial condition samples.

While we acknowledge recent architectural improvements to DeepONets [Krishnapriyan et al., 2021], since our goal is to showcase the effectiveness of symmetries, we deploy MLPs for both networks.

### 3.2 Imposing the Symmetry Criterion

To further inform PINNs about the symmetries of the PDE, we use an additional loss term $\mathcal{L}_{\mathrm{sym}}$. Conveniently, this loss sometimes also contains the PDE loss of Eq. (2).

Recall that the infinitesimal criterion of Theorem 2.2 requires that by acting on a solution $(\mathbf{x}, \mathbf{u}^n)$ in the jet space using the prolonged infinitesimal generator $\mathrm{pr}^{(n)} \mathbf{v}$, the PDE equation should remain satisfied. In simple terms, our symmetry loss encourages the orthogonality of $\mathrm{pr}^{(n)} \mathbf{v}$ and the gradient of $\Delta$ – in other words, infinitesimal symmetry transformations are encouraged to fall on the level-sets of $\Delta$, maintaining $\Delta = 0$. Next, we elaborate on this procedure.

Assume that the Lie algebra of the symmetry group of the $n$-th order PDE, $\Delta$, is spanned by $K$ independent vector fields, $\{\mathbf{v}_1, \ldots, \mathbf{v}_K\}$, where each $\mathbf{v}_k$ is defined as in Eq. (6). As noted earlier, for each $\mathbf{v}_k$, we can obtain their prolongations using automatic differentiation and create a vector of the corresponding coefficients:

$$\mathrm{coef}(\mathrm{pr}^{(n)} \mathbf{v}_k) = \big[\xi_0^k, \ldots \xi_p^k, \phi_0^k, \ldots, \phi_q^k, (\phi_0^{x_0})^k, \ldots, (\phi_q^{x^1, \ldots x^p})^k\big] . \tag{10}$$

This is a vector of infinitesimal transformations in the jet space.[5]

---

[5]Using $\mathrm{coef}$ in the equation above is to differentiate the abstract definition of Eq. (6) and a vector of its coefficients.

We also use the notation $J_\Delta$ for the gradient of $\Delta$ wrt all independent and dependent variables: $J_\Delta = \left( \frac{\partial \Delta}{\partial x^i}, \frac{\partial \Delta}{\partial \mathbf{u}_J^\alpha} \right)$.

The symmetry loss encourages the orthogonality of each of $K$ prolonged vector fields and the gradient vector above on points inside the domain:

$$\mathcal{L}_{\text{sym}} = \sum_{k=1}^{K} (J_\Delta^\top \text{coef}(\text{pr}^{(n)} \mathbf{v}_k))^2 \ . \tag{11}$$

An alternative is to minimize the absolute value of cosine similarity. We found both of these to work well in practice.

Therefore, the total loss we use to train the two networks consists of the PINN loss introduced in Eq. (2) and the symmetry loss: $\mathcal{L} = \alpha \mathcal{L}_{\text{PDE}} + \beta \mathcal{L}_{\text{data-fit}} + \gamma \mathcal{L}_{\text{sym}}$, where $\alpha$, $\beta$ and $\gamma$ are hyperparameters. However, as we see through examples, one or more symmetries of a PDE often simplify to a constant times the $\mathcal{L}_{PDE}$, removing the need to separate treatment of the PDE loss. Algorithm 1 summarizes our training algorithm.

---

**Algorithm 1** PINN with Lie Point Symmetry

**inputs**:
    $\Delta$: the PDE equation of order $n$,
    $(\mathbf{v}_k)_{1:K}$ : infinitesimal generators of symmetries of $\Delta$
    $\mathcal{D} = \left\{ \left( \mathbf{x}_{1:N_r}, (\mathbf{x}, \mathbf{u})_{1:N_l}, \mathbf{u}_{1:N_s} \right) \right\}_{1:N_f}$: dataset for $N_f$ different initial conditions
**init:** initialize parameters $\theta$ of network $u_\theta$
**for** iteration **do**
    Sample from $\mathcal{D}$
    **calculate** $\mathcal{L}_{\text{sym}}$:
        calculate $\mathbf{u}^{(n)}$ using automatic differentiation for $\mathbf{x}_{1:N_r}$
        calculate $\text{coef}(\text{pr}^{(n)} \mathbf{v}_k)$, using $\mathbf{x}_{1:N_r}, (\mathbf{u}^{(n)})_{1:N_r}$, auto diff and Eq. (7)
        calculate $\mathcal{L}_{\text{sym}}$ using $\text{coef}(\text{pr}^{(n)} \mathbf{v}_k)$ and $\Delta$ and equation Eq. (11)
    **calculate** $\mathcal{L}_{\text{PDE}}$:
        using $\Delta, (\mathbf{u}^{(n)})_{1:N_r}$ and equation Eq. (3):
    **calculate** $\mathcal{L}_{\text{data-fit}}$:
        calculate $\hat{\mathbf{u}} = \mathbf{u}_\theta(\mathbf{x}, \mathbf{u}_{1:N_s})$ for $\mathbf{x}_{1:N_l}$
        calculate $\mathcal{L}_{\text{data-fit}}$ using $\mathbf{u}_{1:N_l}$ , $\hat{\mathbf{u}}_{1:N_l}$ and equation Eq. (4)
    $\mathcal{L} = \alpha \mathcal{L}_{\text{PDE}} + \beta \mathcal{L}_{\text{data-fit}} + \gamma \mathcal{L}_{\text{sym}}$
    $\theta \leftarrow \theta - \nabla_\theta \mathcal{L}$
**end for**
**return** $u_\theta$

---

## 4 Experiments

### 4.1 Heat Equation

First, we study the effectiveness of imposing the symmetry constraint on the heat equation, described in Eq. (5). This equation is a simple linear PDE with a rich symmetry group.

**Symmetries.** The following 6-dimensional lie algebra spans the symmetry group of the heat equation:

$$\mathbf{v}_1 = \partial_x \quad \mathbf{v}_3 = \partial_u \qquad \qquad \mathbf{v}_5 = 2\nu t \partial_x - xu \partial_u \tag{12}$$
$$\mathbf{v}_2 = \partial_t \quad \mathbf{v}_4 = x\partial_x + 2t\partial_t \quad \mathbf{v}_6 = 4\nu tx\partial_x - 4\nu t^2 \partial_t - (x^2 + 2\nu t)u\partial_u \ .$$

For example, the infinitesimal generator $v_1$ corresponds to space translation, and $v_5$ to Galilean boost. We refer the reader to Olver [1986] for more details on the derivation.

**Data.** We generate simulated solutions, which we use to test the models' performance. We use $\Omega = [0, L] = [0, 2\pi]$, discretized uniformly into 256 points and assume periodic spatial boundaries. We also use $[0, T] = [0, 16]$, discretized into 100 points. The viscosity coefficient is set to $\nu = 0.01$.

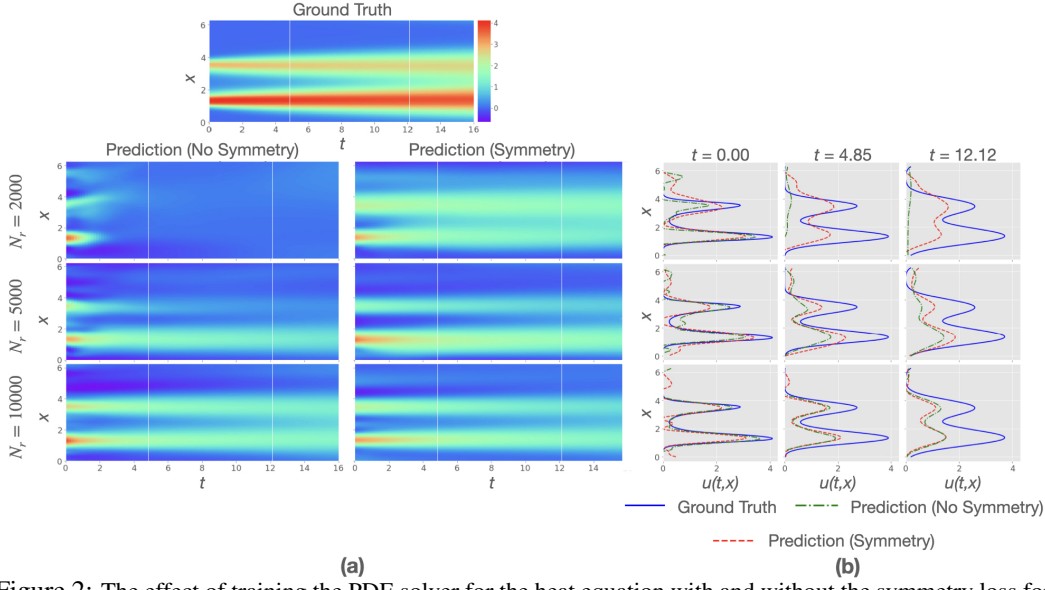

**(a)**                                                                  **(b)**

Figure 2: The effect of training the PDE solver for the heat equation with and without the symmetry loss for one of the PDEs in the test dataset. (a) shows the ground truth solution and the predictions of the two models as the number of samples inside the domain increases from 500 to 2000 and 1000.(b) shows the corresponding predictions and the ground truth solution at different time slices.

Similar to Brandstetter et al. [2022b,a] and Bar-Sinai et al. [2019], we represent the initial condition functions by truncated Fourier series with coefficients $A_k, l_k, \phi_k$ sampled randomly, and $K = 10$:

$$u(t = 0, x) = f(x) = \sum_{i=1}^{K} A_k \sin(2\pi l_k x / L + \phi_k) \,. \tag{13}$$

These functions are sampled at $N_s = 200$ fixed points, which are used as input to $e_{\theta_1}$ in Eq. (9). We also sample a total of $N_l = 300$ points (including the 200 points sampled at $t = 0$), used to impose the data-fit loss, $\mathcal{L}_{\mathrm{data-fit}}$. We also randomly sample 100 and 300 of these initial conditions and use them for the validation and test datasets respectively.

**Training and Experiments.**   The main objective of our experiments is to confirm the hypothesis that training with symmetry loss helps improve the model's prediction capability in a low-data regime. Therefore, we train the model with and without symmetry and evaluate the model's predictions on the test dataset as we increase the number of samples inside the domain, $N_r$. To illustrate the effectiveness of the symmetries in a low-data regime, we use $N_f = 100$ different initial conditions and test the performance as we increase $N_r$ from 500 to 2000 and 10000. We refer to Appendix C for details on the architectures and hyperparameters.

**Results.**   In Table 1, we can see a comparison between the performance of the two models on the test dataset of unobserved initial conditions as $N_r$ increases. We note that when trained with few samples, the model trained with symmetry loss, $\mathcal{L}_{\mathrm{sym}}$ performs significantly better than the baseline model. Fig. 2, also illustrates this point as it shows the performance of both models on a single instance from the test dataset. We note that the improvement in prediction results in the model where symmetry loss is enforced is especially significant at larger values of time, $t$.

We want to highlight an important detail: by using the infinitesimal criterion for enforcing symmetries, not all symmetries of the PDE will help improve the training. There are instances when the gradient of $\Delta$ along the vector field is trivially zero, and in other instances, 'we obtain $c\Delta$ where $c$ is a constant. In the case of the heat equation, only the symmetry transformations from vector fields $v_5$ and $v_6$ provide useful training signals. This means that in our experiments, we can simply eliminate the PDE loss, $\mathcal{L}_{\mathrm{PDE}}$, and only use the symmetry loss, $\mathcal{L}_{\mathrm{sym}}$, in addition to the supervised loss.

| Table 1: The average test set mean-squared error for the Heat equation. | | |
| --- | --- | --- |
| Number of Points $(N_r)$ | No Symmetry | Symmetry |
| 500 | $1.12 \pm 0.58$ | $\mathbf{0.30 \pm 0.15}$ |
| 2000 | $0.36 \pm 0.19$ | $\mathbf{0.24 \pm 0.14}$ |
| 10000 | $0.22 \pm 0.14$ | $\mathbf{0.21 \pm 0.13}$ |

| Table 2: The average test set mean-squared error for Burgers' equation. | | |
| --- | --- | --- |
| Number of Points $(N_r)$ | No Symmetry | Symmetry |
| 5000 | $0.041 \pm 0.042$ | $\mathbf{0.034 \pm 0.039}$ |
| 25000 | $0.030 \pm 0.038$ | $\mathbf{0.017 \pm 0.020}$ |
| 100000 | $0.018 \pm 0.022$ | $\mathbf{0.013 \pm 0.020}$ |

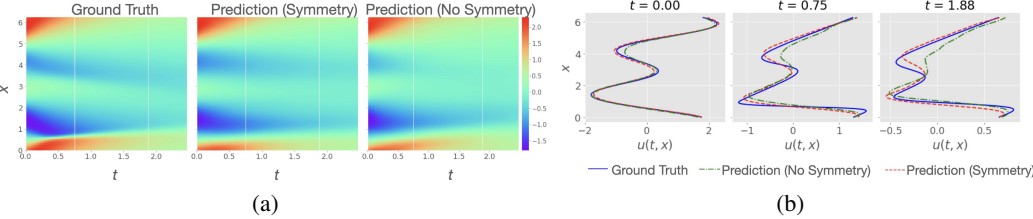

(a)                                                    (b)

Figure 3: (a) predictions of the network trained with a total of 100000 points inside the domain (500 initial conditions and only 200 points for each PDE), with and without the symmetry loss, for Burgers' equation on a test data. (b) corresponding predictions at different time slices showing that the model trained with symmetry loss performs better, especially at larger $t$.

## 4.2 Burgers' Equation

The second PDE we analyze is Burgers' Eq. (14), which combines diffusion (with thermal diffusivity $\nu$) and non-linear advection (wave motion). The nonlinearity of this equation makes it more complex, resulting in shock formation.

$$u_t = \nu u_{xx} - u u_x \tag{14}$$

**Symmetries.** Burgers' equation in the form described in Eq. (14) has a symmetry group spanned by the following 5-dimensional vector space.

$$\begin{aligned} \mathbf{v}_1 &= \partial_x & \mathbf{v}_3 &= t\partial_x + \partial_u & \mathbf{v}_5 &= tx\partial_x + t^2\partial_t - (x - tu)\partial_u \\ \mathbf{v}_2 &= \partial_t & \mathbf{v}_4 &= x\partial_x + 2t\partial t - u\partial_u \end{aligned} \tag{15}$$

However, only the last generator $\mathbf{v}_5$ results in a useful training signal. The first three generators give $\mathcal{L}_{\text{sym}} = 0$ and $\mathbf{v}_4$ gives $\mathcal{L}_{\text{sym}} = c\Delta = c\mathcal{L}_{\text{PDE}}$, for a constant $c$. As with the heat equation experiment, we can eliminate the PDE loss and only use symmetry and supervised losses for training.

**Data.** The data used to evaluate the model is obtained using the Fourier Spectral method with periodic spatial boundaries. Initial conditions are obtained similarly to the heat equation experiment, described in Eq. (13). We use $\nu = 0.1$ as the diffusion coefficient. The domain is $[0, L] = [0, 2\pi]$ and $[0, T] = [0, 2.475]$ discretized uniformly into 256 and 100 points respectively.

**Training and Experiments.** For Burgers' equation, we train the model on datasets of $N_f = 500$ initial conditions and $N_r = 5000, 25000$ and $100000$ samples. We found that for $\mathcal{L}_{\text{sym}}$, cosine similarity works better in this case. The models' architectures are similar to those used for the heat equation described in Appendix C.

**Results.** Table 2 shows the average mean-squared errors on the test dataset for the two models as $N_r$ increases. We can see that, even with one symmetry group useful for training, the model trained with $\mathcal{L}_{\text{sym}}$ performs better. The predictions on a single instance of the test dataset can also be seen in Section 4.2. Again, we see that the symmetry loss especially improves the model's performance for larger values of $t$. See Appendix B for additional plots of the prediction results. We also note that the high standard deviations in Table 2 are because, compared to the heat equation, the behaviour of the solution (specifically shock formation) varies a lot based on the initial conditions.

## Conclusion and Limitations

Lie groups and continuous symmetries are historically rooted in the study of differential equations, yet to this day, their application to Neural PDE solvers has been limited. Our work presents the

foundations for leveraging Lie point symmetry in a large family of Neural PDE solvers that do not require access to accurate simulations. Using available machinery of automatic differentiation, we show that local symmetry constraints can improve PDE solutions found using PINN models.

The method we propose to leverage local symmetries has some limitations: 1) while the Lie point symmetries of important PDEs are well-known, in general, one needs to analytically derive them for a given PDE to use our approach; 2) as we mentioned in Section 4, not all symmetries of the equation will necessarily be useful for constraining the PINN. Fortunately, the usefulness of symmetries is obvious from the corresponding infinitesimal criterion, and one could limit the symmetry loss to useful symmetries; 3) while symmetries can significantly improve performance, based on our empirical observations (see Table 1), one could achieve a similar effect with PINN by increasing the sample size. These limitations motivate our future direction, which builds on our current understanding, to impose symmetry constraints through equivariant architectures.

### Acknowledgments

The authors would like to thank Prakash Panangaden for insightful discussions on symmetries and Lie Group theory. We also thank Oumar Kaba for his helpful feedback on this manuscript. This research is in part supported by Canada CIFAR AI Chair and Microsoft Research. Computational resources are provided by Mila and Digital Research Alliance of Canada.

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

# A   Illustrative Examples

**Example 3** (Obtaining the Prolongation of $SO(2)$). *We can consider If $X \times U = \mathbb{R} \times \mathbb{R}$ and the infinitesimal generator of the 2-dimensional rotation group, $SO(2)$:*

$$\mathbf{v}_{SO(2)} = \xi(x, u)\partial_x + \phi(x, u)\partial_u$$
$$= -u\partial_x + x\partial_u$$

*In this 2-dimensional case, the calculation of the prolonged generator is simple:*

$$\phi^{(x)} = D_x(\phi - \xi u_x) + \xi u_{xx}$$
$$= D_x(x + uu_x) - uu_{xx}$$
$$= (1 + u_x^2 + uu_{xx}) - uu_{xx}$$
$$= 1 + u_x^2$$

*Therefore:*

$$\mathrm{pr}^{(1)}\mathbf{v}_{SO(2)} = -u\partial_x + x\partial_u + (1 + u_x^2)\partial_{u_x}$$

We will work through another example of obtaining the prolongation of an infinitesimal generator of the heat equation:

**Example 4** (Obtaining the Prolongation of an Infinitesimal Generator). *As an example, we will consider $X \times U = \mathbb{R}^2 \times \mathbb{R}$ and the following infinitesimal generator, which is a symmetry of the heat equation:*

$$\mathbf{v} = \xi_1(x, t, u)\partial_x + \xi_2(x, t, u)\partial_t + \phi(x, t, u)\partial_u$$
$$= 2\nu t\partial_x - xu\partial_u$$

*where $x, t$ denote the independent variables, $u$ is the dependent variable and $\nu$ is a positive constant. By the prolongation formula, Eq. (7), the first prolongation in $t$ is given by:*

$$\phi^t = D_t(\phi - \xi_1 u_x - \xi_2 u_t) + \xi_1 u_{xt} + \xi_2 u_{tt}$$
$$= D_t(-xu - 2\nu t u_x) + 2\nu t u_{xt}$$
$$= -xu_t - 2\nu u_x$$

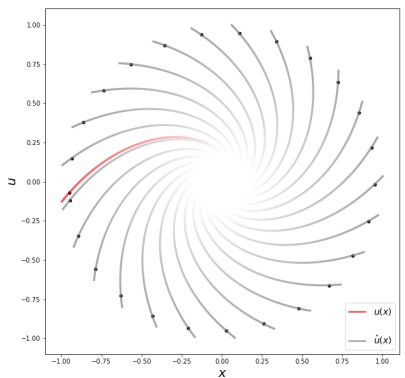

Figure 4: Various solutions of the PDE $\Delta(x, u, u_x) = (u - x)u_x + u + x = 0$ obtained via symmetry transformation (rotation) of a know solution (in red).

As a final illustrative example of the symmetry criterion, we will follow Olver's example below:

**Example 5.** *As an illustrative example of the infinitesimal criterion, we can consider a simple DE:*

$$\Delta(x, u, u_x) = (u - x)u_x + u + x = 0$$

*We can easily see that $SO(2)$ is a symmetry group of this differential equation, using the prolongation of the generator we calculated in Example 3:*

$$\mathrm{pr}^{(1)}\mathbf{v}[\Delta] = -u\Delta_x + x\Delta_u + (1 + u_x^2)\Delta_{u_x}$$
$$= -u(1 - u_x) + x(1 + u_x) + (1 + u_x^2)(u - x)$$
$$= u_x\Delta$$

*Since $\Delta u_x = 0$ when $\Delta = 0$, we can conclude that $SO(2)$ is indeed a symmetry group of the equation. In fact, we can see that it transforms solutions of this differential equation to other solutions in Fig. 4.*

## B  Implementation and Training Details

We model the two networks, $g_{\theta_1}$ and $e_{\theta_2}$ in Eq. (9) with MLPs consisting of 7 hidden layers of width 100. This choice was based on the previous research using PINN and DeepONets for solving Burgers' equation [Wang et al., 2021]. We used elu activation as differentiable activations are required for the PDE loss. The output of the embedding vectors from both networks is 100 dimensional. We used ADAM optimizer with learning rate of $0.001$ for the training and performed early stopping using the validation dataset.

For both the Heat equation and Burgers' equation experiments, we perform hyper-parameter tuning on the coefficients of the loss terms from the set $[0.1, \ldots, 1, \ldots, 10, \ldots, 100, \ldots 200]$. This is done separately for the baseline model and the model trained with symmetry loss, $\mathcal{L}_{\text{sym}}$, as we varied the number of samples, $N_r$. We found that similar to PINNs, the model is sensitive to the weight given to the supervised loss for the initial conditions vs the symmetry/PINN loss. The specific coefficient values for the models trained with and without symmetry loss for the heat equation are in Table 3 and Table 4.

Table 3: The loss coefficients for the model trained with the symmetry loss (i.e. $\mathcal{L} = \beta\mathcal{L}_{\text{sym}} + \gamma\mathcal{L}_{\text{data-fit}}$ for the heat equation for various number of unique points sampled inside the grid for training, $N_r$

.

| loss coefficient | $N_r = 500$ | $N_r = 2000$ | $N_r = 10000$ |
|---|---|---|---|
| $\beta$ | 100 | 80 | 40 |
| $\gamma$ | 20 | 20 | 20 |

Table 4: he loss coefficients for the model trained without the symmetry loss (i.e. $\mathcal{L} = \alpha\mathcal{L}_{\text{PINN}} + \gamma\mathcal{L}_{\text{data-fit}}$ for the heat equation for various number of unique points sampled inside the grid for training, $N_r$

.

| loss coefficient | $N_r = 500$ | $N_r = 2000$ | $N_r = 10000$ |
|---|---|---|---|
| $\alpha$ | 150 | 150 | 130 |
| $\gamma$ | 20 | 20 | 20 |

We also note that for Burgers' equation, we found that cosine similarity for $\mathcal{L}_{\text{sym}}$ works better than the dot product. The results reported in Section 4 use cosine-similarity.

We will make the data and the code available on GitHub.

## C  Additional Results

In the figure below, we can see the behaviour of the two models, trained with and without symmetry loss for Burgers' equation, as we increase the number of training samples. It can be seen that, as expected, in the model trained with $\mathcal{L}_{\text{sym}}$ performs significantly better with low samples inside the domain. The corresponding mean-squared errors are reported in Table 2.

To show the effect of the number of symmetry groups of a PDE in the performance of the model trained with the symmetry loss, we analyzed the performance on (average MSE over the test dataset) as we increased the number of infinitesimal generators that were used to calculate $\mathcal{L}_{\text{sym}}$ for the Heat equation. The two tables below show this effect for two different models that are trained with $N_r = 500$ and $N_r = 10000$ unique points sampled from inside the grid, respectively. As it can be seen, including both of the useful symmetries of the Heat equation leads to the best performance compared to the models that are trained with 1 or 0 infinitesimal generators. Furthermore, as shown in previous results, we can see that the model's overall performance in all 3 cases increases as we

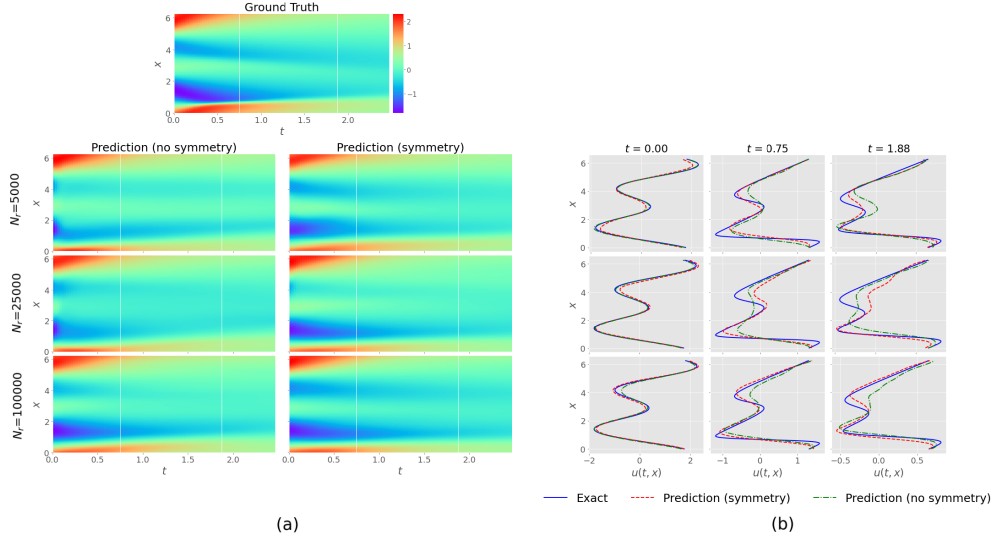

(a)                                                    (b)

Figure 5: The effect of training the PDE solver for the Burgers' equation with and without the symmetry loss for one of the PDEs in the test dataset. (a) shows the ground truth solution and the predictions of the two models as the number of samples inside the domain increases from 5000 to 25000 and 100000.(b) shows the corresponding predictions and the ground truth solution at different time slices.

sample more points inside the grid and that the effect of the symmetry loss is more pronounced in the low-data regime. We also tried training models with and without a symmetry loss for the

Table 5: The average test set mean-squared error for the Heat equation as a function of increasing the number of infinitesimal generators used to compute the symmetry loss. The MSE is reported for models trained with different numbers of unique points sampled from inside the grid. It can be seen that including both of the useful symmetries of the Heat equation leads to the best performance compared to the models that are trained with 1 or 0 infinitesimal generators.

| Number of Symmetries ($K$) | MSE when $N_r = 500$ | MSE when $N_r = 10000$ |
| --- | --- | --- |
| 0 | $0.73 \pm 0.38$ | $0.26 \pm 0.21$ |
| 1 | $0.61 \pm 0.33$ | $0.25 \pm 0.17$ |
| 2 | $\mathbf{0.44 \pm 0.26}$ | $\mathbf{0.20 \pm 0.12}$ |

Heat equation using a modified MLP architecture, as suggested in Wang et al. [2020b]. We found that, unexpectedly, this architecture was performing worse than a simple MLP architecture in our experiments. However, as it can be seen in Table 6, the model trained with the symmetry loss still performs better than the one trained with just the PINN loss.

Table 6: The average test set mean-squared error for the Heat equation as a function of increasing the number of unique points sampled inside the grid. The model architecture is a modified MLP, as suggested in Wang et al. [2020b]. It can be seen that the model trained with the symmetry loss performs betters than that trained without, especially in the low-data regime.

| Number of Points ($N_r$) | No Symmetry | Symmetry |
| --- | --- | --- |
| 500 | $1.15 \pm 0.754$ | $\mathbf{0.859 \pm 0.689}$ |
| 2000 | $0.974 \pm 0.667$ | $\mathbf{0.520 \pm 0.488}$ |
| 10000 | $0.391 \pm 0.577$ | $\mathbf{0.357 \pm 0.363}$ |

