# OpenReview forum: "Lie Point Symmetry and Physics-Informed Networks"
_NeurIPS.cc/2023/Conference — NeurIPS 2023 poster_

### Official Review · Reviewer_KJvK · 2023-07-03

**Soundness:** 3 good
**Presentation:** 2 fair
**Contribution:** 3 good
**Rating:** 6
**Confidence:** 4

**Summary:**

This study proposes a new loss function for PINNs that imposes the symmetry of PDE.


**Strengths:**

PINN loss comes from the equation, but the proposed loss comes from the property of the equation. The idea is insightful.


**Weaknesses:**

The additional loss increases the computational cost. With the same computational budget, one can increase the number of evaluation points (N_r?) instead of introducing the proposed loss. The comparison might not be fair.

The proposed method was only evaluated with the heat equation and Burgers' equation, which are very simple PDEs. Is the symmetry found in more practical and complicated PDEs? Then, is the proposed method useful?

The symbols are not unified. In (7), N_l or N_r? Just after (13), N_l=300 is used for data-fit, but the number of points for data-fit is N_0 in (8).

In Table 1, "1000" might be wrong.


**Questions:**

With the same computational cost, is the proposed method superior to the vanilla PINN?

Is the symmetry found in more practical and complicated PDEs? Then, is the proposed method useful?


**Limitations:**

The proposed method is limited to PDEs with known symmetry. The generality is unknown.

---

> ### Author Rebuttal · Authors · 2023-08-10
>
> We thank the reviewer for their useful feedback! Below we answer their questions and comments.
>
> **R**: The additional loss increases the computational cost. With the same computational budget, one can increase the number of evaluation points ($N_r$?)
>
> **A**: This is true. We have included experiments comparing the effect of the number of evaluation points to that of symmetry loss.
>
> **R**: The proposed method was only evaluated with the heat equation and Burgers' equation, which are very simple PDEs. Is the symmetry found in more practical and complicated PDEs? Then, is the proposed method useful?
>
> **A**: While symmetries are present in more complicated PDEs and generally, one may expect more symmetries in higher dimensions, we had a hard time adapting existing boundary-conditioned PINN models to produce a reasonable performance in these settings so that we can then improve their results using symmetry loss. To see the effect of these symmetry constraints on larger and more complex problems, we believe the symmetry loss should be combined with further innovations in PINNs that can enable its application to high-dimensional PDEs.
>
> **R**: The symbols are not unified. In (7), $N_l$ or $N_r$? Just after (13), $N_l=300$ is used for data-fit, but the number of points for data-fit is $N_0$ in (8).
>
> **A**: Thanks for pointing out the typos! In (7) it should be $N_r$. $N_l$ is used to refer to samples from both initial and boundary conditions ($N_l = N_0 + N_b$). We will make the notation more consistent and clear in the revision.
>
> **R**: In Table 1, "1000" might be wrong.
>
> **A**: Yes, thanks! It should be 10000.

---

> > ### Comment · Reviewer_KJvK · 2023-08-19
> > **Thank you for your response.**
> >
> > Thank you for your response and additional experiments.
> >
> > As I am concerned, simply using much more points leads to a better result. While introducing symmetry is always effective, I would like to see curves of the computational cost vs performance for a fair comparison. In other words, with the same computational budget, is introducing the symmetry loss more effective than increasing the number of points?
> >
> > > Finally, a common question is about the automatic derivation of symmetries from the PDE. This is indeed possible – computational algebraic software can calculate symmetry groups for a given PDE. Symbolic programs in MACSYMA, REDUCE, MAPLE, and Mathematica have been developed to find the equations for the infinitesimal generators.
> >
> > It is not easy to agree on this point without a demonstration. If you could demonstrate your method with such a procedure, I would give it a better score.

---

### Official Review · Reviewer_5AKh · 2023-07-04

**Soundness:** 3 good
**Presentation:** 3 good
**Contribution:** 2 fair
**Rating:** 6
**Confidence:** 3

**Summary:**

In this paper, a method for finding solutions to differential equations that represent physical phenomena using neural networks is considered. In particular, a method that takes symmetry into account is proposed. Specifically, the authors consider an infinitesimal generator that represents the symmetry of the equation, and constrain the prolongation of the solutions to be orthogonal to the isosurface of the solution expressed in Jet space. This is expected to improve the accuracy of the solutions.

**Strengths:**

As far as I know, the use of the symmetry of the model as an infinitesimal generator, rather than in the form of conservation laws, is certainly new. This enables the use of symmetry for equations that are not derived from the variational principle. The proposed method would be reliable in the sense that the method has a theoretical basis. In addition, the paper is clearly written.

**Weaknesses:**

The computation and implementation of symmetries and the constraints on the jet space based on them are considered to be quite difficult. On the other hand, the improvement confirmed by the numerical experimentation is not significant, so the effect is limited compared to the difficulty of implementation.

**Questions:**

(1) Can the computation of symmetry be automated by using computational algebraic software (e.g., Mathematica, Maple, Sigular)?
(2) Although the improvement in numerical experiments is not significant for the cases considered in this paper, can it be effective for systems with many symmetries and conservation laws, such as integrable systems?
(3) Perhaps I am misunderstanding something, but is it not sufficient to simply add some additional constraints? For example, does introducing the higher-order derivative of the loss function as additional cost functions (i.e. Sobolev learning) have a similar effect as the proposed method? I suppose that this can be used as an additional constraint because If the loss function is zero for all x and t, then the derivative of the loss function with respect to x and t should also be zero. In this way, it seems to me, it would be easy to create additional constraints without the laborious symmetry computations.

**Limitations:**

No potential negative societal impact is expected.

---

> ### Author Rebuttal · Authors · 2023-08-10
>
> We thank the reviewer for their insightful comments. Below we respond to their individual questions and comments.
>
> **R**: Can the computation of symmetry be automated by using computational algebraic software (e.g., Mathematica, Maple, Sigular)?
> **A**: Yes, computational algebraic software can be used to calculate symmetry groups for a given PDE. Symbolic programs in MACSYMA, REDUCE, MAPLE, and Mathematica have been developed to find the equations for the infinitesimal generators. For example, the document  (not ours) in the link below is a guide for one such package. We thank the referee for pointing out the fact that we had not mentioned this, and we will clarify the draft.
> (https://docs.google.com/viewer?url=https%3A%2F%2Flibrary.wolfram.com%2Finfocenter%2FID%2F4231%2FYaLie.ps%3Ffile_id%3D3408)
>
> **R**: Although the improvement in numerical experiments is not significant for the cases considered in this paper, can it be effective for systems with many symmetries and conservation laws, such as integrable systems?
>
> **A**: Interesting point! Our hypothesis is that symmetry loss will lead to more improvement for a system with many symmetries. We have added an experiment (see the attached PDF) where we track the improvement due to symmetry loss as we use more symmetries for the heat equation. This experiment shows how increasing the number of infinitesimal generators used in calculating the symmetry loss leads to improved performance. This hypothesis is also confirmed when comparing heat to Burgers’ since we can see that the greater number of symmetries for the heat equation leads to greater improvement in the results.
>
> **R**: Is it not sufficient to simply add some additional constraints (e.g. higher order derivatives of the loss)?
>
> **A**: Additional constraints on higher derivatives of the PINN loss are very different from constraints on derivatives produced through symmetry loss. The former is changing the PINN loss (rather than the solution) for example making the PINN loss smoother, and we’re not sure how it will affect the performance. However, the symmetry loss is explicitly encouraging that certain infinitesimal transformations of the solution found by the network remain solutions to the PDE. We will add a discussion of these points to the revision.

---

> > ### Comment · Reviewer_5AKh · 2023-08-17
> >
> > Thank you for the experimental results on the relationship between the number of symmetries and the performance of the method. Based on these results, it seems that the proposed method is very effective for equations with a large number of symmetries and conservation laws, such as integrable PDEs. In particular, if the method can be combined with computational algebra so that a large number of symmetries can be easily, or automatically, handled, it can be a very powerful method.
> >
> > However, as the number of symmetries in the additional experiment is limited, there remains a concern that as the number of symmetries increases, the performance improvement may saturate. So I will keep my score this time; however, if additional experiments that show that the performance improvement would not saturate are provided, I am happy to increase the score.

---

> > > ### Author Response · Authors · 2023-08-20
> > >
> > > We thank the reviewer for acknowledging the significance of the proposed method. Unfortunately, current PINN models that can take as input different initial conditions fail for higher dimensions. This is due to inherent difficulties in training PINN models, the cost of calculating gradients, and other gradient pathologies that have been studied previously [1]. Given these limitations, it isn’t feasible to showcase the performance of the proposed algorithm for high-dimensional or complex systems. We hope that further developments in PINN model will allow us to exploit their symmetries with the proposed algorithm.
> > >
> > > [1] Sifan Wang, Yujun Teng, and Paris Perdikaris. Understanding and mitigating gradient pathologies in physics-informed neural networks, 2020b.

---

> > > > ### Comment · Reviewer_5AKh · 2023-08-21
> > > >
> > > > Thank you so much for your response. Although not completely, to some extent, my concerns have been addressed and I have increased the score.

---

### Official Review · Reviewer_dief · 2023-07-06

**Soundness:** 3 good
**Presentation:** 1 poor
**Contribution:** 2 fair
**Rating:** 3
**Confidence:** 4

**Summary:**

The work proposes a generic method to incorporate Lie point symmetry into physics-informed neural networks (PINNs) by augmenting loss function. The method leverages automatic differentiation as other PINNs do because the condition for symmetries is written using differentials. The authors demonstrated the model with symmetry gained better performance than one without symmetry.

**Strengths:**

* The work is based on an established mathematical theory found in Olver (1986).
* Once the Lie point symmetries are obtained, it seems quite simple to incorporate them, thanks to automatic differentiation and the PINNs framework.

**Weaknesses:**

* The experiments look insufficient. They compared with and without symmetry but no other state-of-the-art PINN-based models. It makes the contribution of the work look limited because most of the mathematical contributions directly come from Olver (1986), and the results from the symmetry model are not good enough (in particular, Figure 2). They claimed that "our goal is to showcase the effectiveness of using symmetries" (lines 230-231). However, the positive effect of the symmetries is well-known to the community (e.g., Wang et al. 2021a).
* The superiority of incorporating symmetries through loss function is unclear. I suggest the authors include data augmentation methods (e.g., Brandstetter et al. 2022a) and equivariant models (e.g., Wang et al. 2021a).
* The authors claimed, "Our work presents the foundations for leveraging Lie point symmetry in a large family of Neural PDE solvers that do not require access to accurate simulations" (lines 316-318). However, another part says, "The data for Burgers' equation is obtained using the Fourier Spectral method" (line 299). It makes the reviewer confused about whether the method uses simulation data as supervisory signals.
* The contribution of the paper is not clear enough. Is it a new theorem proved, an undiscovered observation found, a state-of-the-art performance, or a novel problem setting? The reviewer recommends the authors clearly state the contribution of the work and show the supporting facts.
* The mathematical presentation lacks correctness and clarity. The reviewer found no clear definition of "Lie point symmetry." The definition of $u^{(n)}$ lacks $u$ (see Olver (1986) p.97). It may be by mistake; however, this is an essential part of the theory. Thus the reviewer recommends the authors carefully re-check the manuscript for more mathematical correctness. For instance, if there is no $u$ in $u^{(n)}$, one cannot express the advection term of the Burgers' equation in the jet space.

Minor points:
* Please add an explanation about $N_s$.
* The paper uses "viscosity" for the heat equation, which seems uncommon. It could be called the diffusion coefficient.
* In the first equation of Equation (1), $t$ should be defined in an open set (see, e.g., Jürgen Jost "Partial Differential Equations ThirdEdition" (2012) Chapter 1). That's why we need the initial condition.

**Questions:**

* How long do the training and prediction take, respectively?
* What is the meaning of "orthogonality" in the paper, e.g. "our symmetry loss encourages the orthogonality of $\mathrm{pr}^{(n)}v$ and the gradient of $\Delta$" (line 237)? The condition contains PDE itself, so the reviewer guesses the use of the terminology could require some explanation.

**Limitations:**

The limitations are stated sufficiently in the paper.

---

> ### Author Rebuttal · Authors · 2023-08-10
>
> We thank the reviewer for their feedback! Below we respond to individual questions and comments.
>
> **R**: … It makes the contribution of the work look limited because most of the mathematical contributions directly come from Olver (1986), and the results from the symmetry model are not good enough (in particular, Figure 2). They claimed that "our goal is to showcase the effectiveness of using symmetries" (lines 230-231). However, the positive effect of the symmetries is well-known to the community (e.g., Wang et al. 2021a).
>
> **A**: While the theory of Lie point symmetry in PDE’s is an established area, our novel contribution is the methodology for incorporating this symmetry and, in particular, the design of the symmetry loss for PINNs. To our knowledge, our’s is the first work showing the effectiveness of using Lie point symmetries for PINN (Wang et al.22a, while quite relevant, is not concerned with PINNs).
>
> **R**: The superiority of incorporating symmetries through loss function is unclear. I suggest the authors include data augmentation methods (e.g., Brandstetter et al. 2022a) and equivariant models (e.g., Wang et al. 2021a).
>
> **A**: Data augmentation is not possible for PINN models as they are trained directly with the PDE equation – i.e., there is no ground truth training data in PINN to augment. Therefore, one could simply increase or “augment” the number of sampling points, and our experiments compare the effect of sample size and symmetry. To the best of our knowledge, there is no “equivariant model” for PINN. In general, we expect comparison to other neural solvers that are not PINN to significantly favour those solvers that rely on the underlying true PDE for learning. However, they also suffer from the same issue.
>
> **R**: The authors claimed, "Our work presents the foundations for leveraging Lie point symmetry in a large family of Neural PDE solvers that do not require access to accurate simulations" (lines 316-318). However, another part says, "The data for Burgers' equation is obtained using the Fourier Spectral method" (line 299). It makes the reviewer confused about whether the method uses simulation data as supervisory signals.
>
> **A**: Note that Tte data (solution) is generated only for evaluation purposes, not for training. For evaluation, we report the MSE with respect to the ground truth solution. We will make this point clear in the revision.
>
> **R**: The contribution of the paper is not clear enough. Is it a new theorem proved, an undiscovered observation found, a state-of-the-art performance, or a novel problem setting? The reviewer recommends the authors clearly state the contribution of the work and show the supporting facts.
>
> **A**: Main contributions are the following; we will make them explicit in the revision:
> Methodology to calculate the group action on the jet space of the PDE using automatic differentiation.
> Equation for a symmetry loss that enforces arbitrary Lie-point symmetry of the PDE in PINN models.
> Demonstrate the effectiveness of symmetry relative to increasing the number of sampling points in PINNs.
>
> **R**: Please add an explanation about N_s
>
> **A**: Sorry for this confusion in the notation. We will change N_s to N_0 to be consistent with the notation originally introduced.
>
> **R**: The paper uses "viscosity" for the heat equation, which seems uncommon. It could be called the diffusion coefficient.
>
> **A**: Yes, we will change this.
>
> **R**: In the first equation of Equation (1), t should be defined in an open set
>
> **A**: Thanks for pointing out this typo. It will be fixed in the revision.

---

> > ### Comment · Reviewer_dief · 2023-08-16
> >
> > The reviewer appreciates the responses made by the authors. Now it is clear that the method uses no training data and incorporates symmetries behind PDE, which is new. However, the contribution of the paper does not seem sufficient to be accepted at the conference in terms of methodology and experimental significance. The method is a straightforward implementation of Olver (1986) into PINN, and empirical results showed no significance or only marginal improvement. Therefore, the reviewer keeps the score unchanged.

---

> > > ### Author Response · Authors · 2023-08-20
> > >
> > > We thank the reviewer for their response. We emphasize that while developing the theory of symmetry groups of PDEs indeed exists in the mathematical literature, using them to enforce symmetries in neural PDE solvers is a novel contribution. More explicitly, the contributions are listed below, and we will include this list in the revision:
> > >
> > > 1. Methodology to calculate the group action on the jet space of the PDE using automatic differentiation.
> > > 2. Equation for a symmetry loss that enforces arbitrary Lie-point symmetry of the PDE in PINN models.
> > > 3. Demonstrate the effectiveness of symmetry relative to increasing the number of sampling points in PINNs.

---

### Official Review · Reviewer_qo6e · 2023-07-13

**Soundness:** 3 good
**Presentation:** 2 fair
**Contribution:** 2 fair
**Rating:** 4
**Confidence:** 2

**Summary:**

This paper proposes to enhance Physics-Informed Neural Networks (PINNs) by incorporating local Lie-point symmetry into them. It is achieved by introducing an additional symmetry loss term, which requires analytic computation of the PDE’s symmetries.
This loss term is designed to encourage orthogonality between the PDE equation and the different symmetry transformations.
Evaluation is performed on 1D heat and Burgers equations.They show that the incorporation of symmetry gives better performance, especially when the number of sampled points is low.
Being unfamiliar with Lie theory, I found the theoretical to be quite challenging.
Overall, I found the paper interesting, but a bit short on the experimental side.

**Strengths:**

- The proposed approach is theoretically motivated, and leads to good performance compared to vanilla PINN
- It doesn’t need many modifications to the vanilla PINN to implement

**Weaknesses:**

- The proposed method needs to compute the symmetries analytically first, before integrating them into the loss function. Would there be a way to automate this part?
- In the considered examples, the better performance obtained with the additional loss terms could have also been achieved by denser sampling, as has been noted. What

**Questions:**

- How does the number of symmetry constraints scale wrt to the dimensionality of the considered PDE?
- Could this symmetry loss term be incorporated to other PINN variants too?

**Limitations:**

- The experiments are only showcased on 1D PDEs

---

> ### Author Rebuttal · Authors · 2023-08-10
>
> We thank the reviewer for their valuable feedback! Below we respond to their questions and comments.
>
> **R**: The proposed method needs to compute the symmetries analytically first, before integrating them into the loss function. Would there be a way to automate this part?
>
> **A**: Yes, computational algebraic software can be used to calculate symmetry groups for a given PDE. Symbolic programs in MACSYMA, REDUCE, MAPLE, and Mathematica have been developed to find the determining equations for the infinitesimal generators. For example, the document (not ours) in the link below is a guide for one such package.
> (https://docs.google.com/viewer?url=https%3A%2F%2Flibrary.wolfram.com%2Finfocenter%2FID%2F4231%2FYaLie.ps%3Ffile_id%3D3408)
>
> **R**: In the considered examples, the better performance obtained with the additional loss terms could have also been achieved by denser sampling, as has been noted. What
>
> **A**: We agree that increasing sample size, similar to equivariance/symmetry, leads to better generalization. We show the effect of both symmetry and increased sample size for both heat and Burgers' equations. Please note that the question was incomplete.
>
>
> **R**: How does the number of symmetry constraints scale wrt to the dimensionality of the considered PDE?
>
> **A**: The number of symmetry constraints corresponds to the number of infinitesimal generators of the symmetry group of the PDE. While this number depends on the PDE equation itself, one generally expects that in higher dimensional PDE, the existence of certain symmetries (e.g., Euclidean symmetry) leads to an increased number of these infinitesimal generators.
>
>
> **R**: Could this symmetry loss term be incorporated to other PINN variants too?
>
> **A**: Yes, this loss term can be used in any PINN model. As an example, we have added an experiment for the Heat equation with a modified MLP model introduced in [1]. We also tried the causal training suggested in [2]. However, in the operator learning setting that we were trying (i.e. using DeepONet architecture to handle different initial conditions), we did not get improvement to the results as only the models trained with a small $\epsilon$ in the PDE equation ( slope of temporal weights) led to reasonable predictions.
>
>
>
> [1] Sifan Wang, Yujun Teng, and Paris Perdikaris. Understanding and mitigating gradient pathologies in physics-informed neural networks, 2020b.
>
> [2] Sifan Wang, Shyam Sankaran, & Paris Perdikaris. (2022). Respecting causality is all you need for training physics-informed neural networks.

---

> > ### Comment · Reviewer_qo6e · 2023-08-21
> >
> > Thank you for addressing my questions. However, due to the limited numerical experiments, I will keep my rating

---

### Official Review · Reviewer_vAL8 · 2023-07-25

**Soundness:** 3 good
**Presentation:** 2 fair
**Contribution:** 3 good
**Rating:** 6
**Confidence:** 3

**Summary:**

The paper proposes adding a custom loss function to training physics-informed neural networks (PINNs) to force symmetry requirements when learning solutions to PDEs. A given partial differential equation (PDE) is associated with a Lie group that acts on the space of solutions of the PDE that leaves this space invariant. The custom loss penalizes non-orthogonality between the gradient from the PDE and the symmetry constraint. This forces the updates to the solution to be on the level-set of group invariant solution. The loss is computed using the action of the Lie algebra generators on the jet space of the solutions.

The authors show that the additional loss leads to better data efficiency and accuracy in two test case of  Heat Equation and Burgers’ Equation.

**Strengths:**

- The idea of adding symmetry constraints to neural networks PDE solvers is a very interesting contribution to the field.
- The execution is novel and makes clever use of Lie algebra theory on PDEs.
- The exact form of the loss function is clearly motivated.
- The experimental results are compelling and convincing.

**Weaknesses:**

- The paper would benefit significantly from a more thorough ablation study.  Specifically, an exploration of the effects of incrementally adjusting the relative weighting of the different loss terms may offer crucial insights. It would be nice to analyse the data efficieny in a power low for example, as a function of the loss weights.
- The discussion surrounding the two types of loss used in this study (cosine and orthogonality) is not fully clear. While the authors do mention these losses, there is insufficient detail regarding their selection,.
- The mathematical notations of the paper are not clear. Overall, I've found the paper quite hard to parse. For example, the notations $\text{pr}^{(n)}[\Delta]$ is not really explained and I could only guess what it means. The Lie algebras are introduced suddently while not referred before. Please re write the introduction of projector using the Lie algebras.
- There is an absence of any comparison with baseline models using data augmentation techniques or fully equivariant architecture in term of numerical results.

**Questions:**

- See my point on the power law before. I would plot different power laws based on different triplets of loss weights and show the coefficients.
- Please compare to other baselines (including neural network operators and other PINNs) regarding numerical values. There are better ways to compare methods than eyeballing the graphs in Fig. 2 and Fig. 3. In particular because the errors might be very frame dependent. I would give a numerical error by averaging several runs via several seeds over several frames. In the current form, I am unable to fully appreciate the performance to the method. If this point is addressed correctly, I am open to increasing my mark.

**Limitations:**

Limitations have been well discussed.

---

> ### Author Rebuttal · Authors · 2023-08-10
>
> We thank the reviewer for their constructive feedback. Below we respond to the questions and concerns raised in the review.
>
> **R**:  The paper would benefit significantly from a more thorough ablation study. Specifically, an exploration of the effects of incrementally adjusting the relative weighting of the different loss terms may offer crucial insights. It would be nice to analyze the data efficiency in a power low, for example, as a function of the loss weights.
>
> **A**:  We agree with the reviewer that more experiments can always be more helpful. We will elaborate on the following empirical observation that suggests the proposed ablation on loss terms may not add much: We observe that for different problems using the same ratio for PINN and symmetry loss performs well, and further adjustment leads to little change in performance. In contrast, the initial condition matching loss is sensitive (as also seen in prior works on PINN), and therefore we treat it as a hyper-parameter. We believe the suggested ablation will reflect these findings, but we are open to running the experiments if the reviewer still finds it useful.
>
> **R**: There is an absence of any comparison with baseline models using data augmentation techniques or fully equivariant architecture in terms of numerical results.
>
> **A**: Since PINN models are not trained on any dataset, one cannot compare them to data augmentation. This is in contrast to neural operator methods, where data augmentation makes sense. The closest one could get to data augmentation is to add points to PINN loss corresponding to symmetry transformations. But since the value of the dependent variable is unobserved, it cannot be transformed, and the whole scheme simply involves increasing the points for the PINN loss. We do have such experiments. Additionally, to the best of our knowledge, there is no equivariant architecture for PINN model, in part due to the form of the action of the Lie point symmetries on the PDE.
>
>
> **R**: The mathematical notations of the paper are not clear. Overall, I've found the paper quite hard to parse. For example, the notation is not really explained, and I could only guess what it means. The Lie algebras are introduced suddently while not referred before. Please re write the introduction of projector using the Lie algebras.
>
> **A**: We will clarify the specific examples pointed out by the reviewer. The topic is mathematically challenging, and we use a significant portion of the background section introducing many ideas with a running example.
>
> **R**: Please compare to other baselines (including neural network operators and other PINNs)
>
> **A**: Due to access to ground truth data during training, Neural operators generally perform better than PINNs. Since we are improving PINN using symmetry loss, the vanilla PINN model seems like the right baseline. Our objective here is to show that the PINN model at large can be improved through the use of symmetries. To further strengthen the point, we performed new experiments comparing the variant of [X] with and without symmetry. Here again, we see the positive effect of symmetry loss, although we do not see an improvement in general performance compared to vanilla MLP.
>
> **R**: regarding numerical values. [...] In particular because the errors might be very frame dependent. I would give a numerical error by averaging several runs via several seeds over several frames. [...] If this point is addressed correctly, I am open to increasing my mark.
>
> **A**: Results in Tables 1 and 2 are consistent with the reviewer’s suggestions. They are produced by averaging the error over multiple initial conditions, which is the main source of variance. The addition of multiple seeds for the same initial condition does not significantly change the results. Images are only provided for qualitative comparison.

---

> > ### Comment · Reviewer_vAL8 · 2023-08-14
> >
> > Thank you for your answers.
> >
> > ### On the ablation
> > When a new method includes free parameters, it is essential to discuss how they have been chosen and what values were settled upon, even if the impact is found to be minimal. I was unable to locate the final values of each coefficient clearly. You refer to Appendix C (line 275) in the text, but this does not cover that specific detail. Since this new loss is the main contribution of the paper, I am requesting greater clarity on this matter.
> >
> > ### On tables
> > I acknowledge Tables 1 and 2, but I find that there is no clear description of what they exactly represent. I have not seen any explanation of how the test set was selected, so I cannot precisely determine what these numbers mean. I encourage the authors to provide a more rigorous description of their experiments and datasets in the Appendix.
> >
> > ### On the presentation
> > I recognize the challenge of presenting a new (for most of the machine learning community) mathematical concept within the constraints of a short paper. However, complex material can be well presented by being succinct and consistent with notations, something that is not always evident in your paper. For example, you begin by introducing the parametrization of Lie groups by one-parameter subgroups, then proceed to discuss the infinitesimal generators of these subgroups. Later in the results, you start using the term "Lie algebras" without having previously mentioned it. It would be more coherent to introduce cleanly what a Lie algebra is, explain the exponential map, and maintain consistency thereafter. In another instance, you first introduce the prolongation of the action of the Lie group on the jet space and then use the same notation to denote the prolongation of the action of the Lie algebra. This could lead to confusion, and I suggest making the notation and explanation more transparent.
> >
> >
> > Overall I will keep my score and encourage the authors to improve the consistency and clarity of their exposition.

---

> > > ### Author Response · Authors · 2023-08-20
> > >
> > > We thank the reviewer for their answers and especially for pointing out how the presentation of the paper can be improved!
> > >
> > > **On the ablation:**
> > >
> > > We performed a hyperparameter search for the coefficients of the loss terms for both models. We found that, similar to PINNs, the model is sensitive to the weight given to the supervised loss for the initial conditions vs the symmetry/PINN loss.
> > > The specific coefficient values for the model trained with symmetry loss are:
> > >
> > > $\beta$=20, $\gamma$=100 for Nr= 500
> > >
> > > $\beta$=20, $\gamma$=80 for Nr= 2000
> > >
> > > $\beta$=20, $\gamma$=40 for Nr= 10000
> > >
> > > The specific coefficient values for the model trained without symmetry loss are:
> > > $\alpha$=150, $\beta$=20 for Nr= 500
> > >
> > >  $\alpha$=150, $\beta$=20 for Nr= 2000
> > >
> > >  $\alpha$=130, $\beta$=20 for Nr= 10000
> > >
> > > We will make sure to include these in the appendix.
> > >
> > > **On tables:**
> > >
> > > We have referred to the tables in the Results section for both of the experiments and described their significance, mentioning how they provide evidence that symmetry loss is most useful in low-data regimes. We failed to mention that the test set of 300 initial conditions was selected randomly from the full dataset, and we will mention this in the revision.
> > >
> > > **On the presentation:**
> > >
> > > We appreciate the reviewer’s constructive criticisms, especially on how to improve the quality of the presentation of the paper. They will serve to improve the overall quality of our paper in the revision. We believe addressing the two points raised is rather straightforward and would not require a major change.

---

### Author Rebuttal · Authors · 2023-08-10

We thank the reviewers for their insightful and constructive feedback! We are happy to see that they found the idea to be motivated (R-qo6e, R-vAL8), insightful (R-KJvK) and novel (R-vAL8, R-5AKh), the empirical results to be compelling (R-vAL8), and the presentations of the paper to be clear (R-5AKh). Below we address individual questions and concerns raised in the reviews.

One common concern was in using alternative baselines. In the attached PDF, Table 4 presents new results comparing the effect of symmetry for different sample sizes using another architecture for PINN that uses a gating mechanism [1]. We do not observe any significant improvements over the vanilla MLP with or without symmetry loss. However, we do observe consistent improvement with the addition of symmetry loss to the PINN loss, especially in a low data regime. We also experimented with Causal PINN [2], but were did not see any improvements with or without symmetry loss over vanilla PINN loss. Another suggested baseline is data augmentation; Since PINN model is not trained on any dataset, we cannot perform data augmentation. One could simply increase the number of training points, and we do have experiments comparing the effect of symmetry loss with that of increasing data points (note that we cannot even “transform” the training points to perform augmentation since the symmetry also acts on the unknown dependent variable.) Yet another family of suggested baselines is Neural Operator-based methods. We expect Neural Operators to generally outperform PINNs, due to their advantage of training on exact solutions. Note that all other symmetry-based neural solvers cited by the reviewers (and in our paper), for example [3], are either Neural Operators or otherwise require exact solutions for training; therefore, they are not comparable to our PINN-based approach. We are unaware of any symmetry-based improvements for PINN.

Another common question was regarding the effect of the number of PDE symmetries on the effectiveness of symmetry loss. Table 3 reports the result of a new ablation, in which we increase the number of symmetries of the same equation used in the symmetry loss. We observe consistent improvement as we incorporate more symmetries.

Some reviewers have raised concerns about the technical difficulty of the material, as it affects readability. We appreciate this difficulty, and our solution to make the content palatable to ML community is to make the paper as self-contained as possible by providing a comprehensive (less than 3 pages) background on Lie point symmetries and using simple running examples and figures. However, some level of difficulty remains inevitable due to the technical nature of the topic.

Finally, a common question is about the automatic derivation of symmetries from the PDE. This is indeed possible – computational algebraic software can calculate symmetry groups for a given PDE. Symbolic programs in MACSYMA, REDUCE, MAPLE, and Mathematica have been developed to find the equations for the infinitesimal generators.

We will clarify these points and include the new results in the revision.


[1] Sifan Wang, Yujun Teng, and Paris Perdikaris. Understanding and mitigating gradient pathologies in physics-informed neural networks, 2020b.

[2] Sifan Wang, Shyam Sankaran, & Paris Perdikaris. (2022). Respecting causality is all you need for training physics-informed neural networks.

[3] Rui Wang, Robin Walters, and Rose Yu. Incorporating symmetry into deep dynamics models for improved generalization, 2021a.

---

### Decision · Program_Chairs · 2023-09-21

**Decision:**

Accept (poster)

**Comment:**

During the initial phase of reviewing, the submission received mixed scores.  Some of the concerns raised related to the clarity of presentation, which have been addressed by the rebuttal. One of the referees comments on the novelty of the work as the symmetries are already known. However, the incorporation of the symmetries in the training of PINNs is a sufficiently novel contribution. There still remain concerns about limited experimental validation. The submission would indeed be strengthened significantly by taking into account the suggestions of the referees. However, there is sufficient merit in the submission to be accepted.